# Cytokines Differently Define the Immunomodulation of Mesenchymal Stem Cells from the Periodontal Ligament

**DOI:** 10.3390/cells9051222

**Published:** 2020-05-14

**Authors:** Christian Behm, Alice Blufstein, Johannes Gahn, Michael Nemec, Andreas Moritz, Xiaohui Rausch-Fan, Oleh Andrukhov

**Affiliations:** 1Department of Conservative Dentistry and Periodontology, University Clinic of Dentistry, Medical University of Vienna, 1090 Vienna, Austria; christian.behm@meduniwien.ac.at (C.B.); alice.blufstein@meduniwien.ac.at (A.B.); johannes.gahn@gmail.com (J.G.); andreas.moritz@meduniwien.ac.at (A.M.); xiaohui.rausch-fan@meduniwien.ac.at (X.R.-F.); 2Department of Orthodontics, University Clinic of Dentistry, Medical University of Vienna, 1090 Vienna, Austria; michael.nemec@meduniwien.ac.at

**Keywords:** mesenchymal stem cells, periodontal ligament, immunomodulation, cytokines, CD4-positive T-lymphocytes

## Abstract

Human periodontal ligament stem cells (hPDLSCs) play an important role in periodontal tissue homeostasis and regeneration. The function of these cells in vivo depends largely on their immunomodulatory ability, which is reciprocally regulated by immune cells via cytokines, particularly interferon (IFN)-γ, tumor necrosis factor (TNF)-α, and interleukin (IL)-1β. Different cytokines activate distinct signaling pathways and might differently affect immunomodulatory activities of hPDLSCs. This study directly compared the effect of IFN-γ, TNF-α, or IL-1β treated primary hPDLSCs on allogenic CD4^+^ T lymphocyte proliferation and apoptosis in an indirect co-culture model. The effects of IFN-γ, TNF-α, and IL-1β on the expression of specific immunomodulatory factors such as intoleamine-2,3-dioxygenase-1 (IDO-1), prostaglandin E_2_ (PGE_2_), and programmed cell death 1 ligand 1 (PD-L1) and ligand 2 (PD-L2) in hPDLSCs were compared. The contribution of different immunomodulatory mediators to the immunomodulatory effects of hPDLSCs in the indirect co-culture experiments was assessed using specific inhibitors. Proliferation of CD4^+^ T lymphocytes was inhibited by hPDLSCs, and this effect was strongly enhanced by IFN-γ and IL-1β but not by TNF-α. Apoptosis of CD4^+^ T lymphocytes was decreased by hPDLSCs per se. This effect was counteracted by IFN-γ or IL-1β. Additionally, IFN-γ, TNF-α, and IL-1β differently regulated all investigated immunomediators in hPDLSCs. Pharmacological inhibition of immunomediators showed that their contribution in regulating CD4^+^ T lymphocytes depends on the cytokine milieu. Our data indicate that inflammatory cytokines activate specific immunomodulatory mechanisms in hPDLSCs and the expression of particular immunomodulatory factors, which underlies a complex reciprocal interaction between hPDLSCs and CD4^+^ T lymphocytes.

## 1. Introduction

Human mesenchymal stem cells (MSCs) are multipotent, non-hematopoietic progenitor cells having self-renewal potential [1], expressing specific surface markers, and possessing a multilineage differentiation potential in vitro [2]. Initially found in bone marrow [3], MSCs reside in various tissues of the human body [4,5]. In 2004, Seo et al. first isolated a heterogenous population of MSCs from the periodontal ligament (hPDLSCs) [6], a highly specialized connective tissue surrounding the tooth’s root, linking it to the alveolar bone [7]. Quiescent undifferentiated hPDLSCs reside in the perivascular niche of the periodontal ligament [8,9] and are homed to inflamed or injured periodontal tissue by sensing specific chemoattractant stimuli. At the injury site, hPDLSCs participate in regulating periodontal tissue regeneration, tissue homeostasis, and local inflammatory processes [4,10,11].

Similarly to other MSCs, hPDLSCs exert mainly immunosuppressive effects and influence different immune cells, such as inhibiting T lymphocyte proliferation and influencing T lymphocyte apoptosis [4,5]. Immunomodulation is currently considered as the major mechanism of MSCs’ therapeutic effect, since differentiation ability of transplanted MSCs in vivo is limited [5]. The most important factors involved in the immunomodulatory function of hPDLSCs are indoleamine-2,3-dioxygenase 1 (IDO-1), prostaglandin E2 (PGE2), tumor necrosis factor-inducible gene 6 protein (TSG-6), programmed cell death 1 ligand 1 (PD-L1), and programmed cell death 1 ligand 2 (PD-L2) [4,12].

The immunomodulatory activity is usually low in resting hPDLSCs and is enhanced by environmental factors, first of all by inflammatory cytokines produced by activated immune cells [13]. Hence, there is a bidirectional interaction between MSCs and immune cells, leading mainly to an immunosuppressive MSC phenotype, which dampens excessive local immune responses [5,14]. The most important inflammatory cytokines affecting MSCs are interferon (IFN)-γ, tumor necrosis factor (TNF)-α, and interleukin (IL)-1β [13,15].

Although the role of inflammatory mediators in the activation of immunomodulatory properties in MSCs is well recognized [4], the contribution of specific cytokines is rather poorly known. Several studies already recognized the variable effects of IFN-γ, TNF-α, and IL-1β on the expression of certain immunomediators in MSC-like cells [16,17,18]. However, to date, the effect of IFN-γ, TNF-α, and IL-1β on the immunomodulatory activities of hPDLSCs has not been directly compared.

Therefore, the main aim of the present study was to directly compare the effects of hPDLSCs on the proliferation and the apoptosis of allogenic CD4^+^ T lymphocytes in the presence of different inflammatory cytokines using an indirect in vitro co-culture model. Particularly, we investigated the effect of IFN-γ, TNF-α, and IL-1β on the ability of hPDLSCs to modulate allogenic CD4^+^ T lymphocytes, since these three cytokines activate different signaling pathways and consequently might differently affect immunomodulatory activities of hPDLSCs. Hence, we further directly compared the influence of IFN-γ, TNF-α, and IL-1β on the expression of IDO-1, PD-L1, PD-L2, and prostaglandin-endoperoxide synthase 2 (PTGS-2) in hPDLSCs in vitro. Additionally, to verify the role of IDO-1, PD-L1, and PTGS-2 in hPDLSCs’ caused effects on CD4^+^ T lymphocytes under different microenvironmental conditions, these immunomediators were inhibited pharmacologically in indirect co-culture experiments. The results of this study highlight that immunomodulation by hPDLSCs is strongly affected by the local microenvironment. Depending on the presence of the cytokine type, specific immunomodulatory activities of hPDLSCs are boosted, which differently influence CD4^+^ T lymphocyte proliferation.

## 2. Materials and Methods

### 2.1. Ethics

The whole study protocol, including the isolation of primary hPDLSCs from patients and CD4^+^ T lymphocyte isolation from volunteers’ whole blood, was approved by the Ethics Committee of the Medical University of Vienna (EK Nr. 1694/2015, extended 2019). The study was performed according to the Declaration of Helsinki and to the Good Scientific Practice Guidelines of the Medical University of Vienna.

### 2.2. hPDLSCs Culture and Verification of MSC Surface Marker Expression

Primary hPDLSCs were isolated from the mid-third surface of the third molars’ roots, which were extracted due to orthodontic reasons from periodontal healthy individuals aged between 18 and 30 years. Before the surgical procedure, the patients gave their informed written consent. Isolated hPDLSCs were cultured under humidified conditions in Dulbecco’s modified Eagles Medium (DMEM, Sigma-Aldrich, St. Louis, USA), which was supplemented with 10% fetal bovine serum (FBS, Gibco, Carlsbad, USA), 50 µg/ml streptomycin (S, Gibco, Carlsbad, USA), and 100 U/ml penicillin (P, Carlsbad, USA). The stemness of isolated hPDLSCs was verified by analyzing hematopoietic and mesenchymal stem/-stromal cells surface marker expression, as described in our previous study [19]. Additionally, cell suspensions were stained using phycoerythrin (PE)-conjugated mouse anti-human CD73 antibody (eBioscience, San Diego, USA). 

### 2.3. CD4^+^ T Lymphocyte Isolation

Human whole blood was collected from one single volunteer, using the lithium- and heparin containing VACUETTE® blood collection system (Greiner Bio-one, Kremsmünster, Austria). In order to concentrate solely on the variable effects of hPDLSCs from different individuals and to reduce CD4^+^ T lymphocyte donor variability, CD4^+^ T lymphocytes were isolated from the same volunteer throughout the study. Whole blood was diluted 1:1 with Hank’s Balanced Salt Solution (HBSS, Life Technologies, Carlsbad, USA), which was followed by Ficoll-Paque (Ge Healthcare, Chicago, USA) density gradient centrifugation. From harvested allogenic peripheral blood mononuclear cells (PBMCs), CD4^+^ T lymphocytes were isolated by immunomagnetic negative selection using MagniSortTM Human CD4^+^ T cell enrichment kit (Invitrogen, Carlsbad, USA).

### 2.4. Experimental Protocols

#### 2.4.1. hPDLSCs Treatment

24 h after seeding an appropriate number of hPDLSCs in 6-well plates, these cells were stimulated with either 5 ng/ml IL-1β [18] or 10 ng/ml TNF-α [20] or 100 ng/ml IFN-γ [21] (all from Invivogen, San Diego, USA) in FBS-free DMEM for all performed experiments. After 48 h incubation, hPDLSCs were proceeded as indicated below. hPDLSCs were additionally treated with either 50 µM PF-06840003 (IDO-1 inhibitor) or 1 µM BMS202 (PD-1/PD-L1 inhibitor) [22] or 1 µM Celecoxib (PTGS-2 inhibitor) [23] before and during indirect co-culture (all from Selleck Chemicals, Houston, USA). PF-06840003 concentration was chosen due to its effect on IDO-1 enzymatic activity in hPDLSCs, presented in Appendix A. 

#### 2.4.2. hPDLSCs/CD4^+^ T Cell Indirect Co-Culture

Primary hPDLSCs were seeded in 6-well plates at a density of 2.5 × 10^5^ cells per well in 3 ml DMEM (Sigma-Aldrich, St. Louis, USA) containing 10% FBS (Gibco, Carlsbad, USA) and 1% P/S (Gibco, Carlsbad, USA). After 24 h, hPDLSCs were stimulated as indicated above. After 48 h, medium was changed to RPMI-1640 (Sigma-Aldrich, St. Louis, USA) containing 10% FBS and 1% P/S. Transwell (TC) inserts (0.4 µm pore size, Sarstedt, Nürnbrecht, Germany) containing 1 × 10^6^ allogenic CD4+ T cells were inserted into each well. CD4^+^ T lymphocyte proliferation was induced by 10 µg/ml phytohemagglutinin-L (PHA-L, eBioscience, San Diego, USA) in the presence or the absence of different cytokines. After five days, CD4^+^ T lymphocytes proliferation and apoptosis were analyzed by flow cytometry. CD4^+^ T lymphocytes cultured in the absence of hPDLSCs served as reference.

Additionally, hPDLSCs in co-culture were treated with pharmacological inhibitors against IDO-1, PD-L1, and PTGS-2, as mentioned above. We aimed to clarify the role of these immunomediators on hPDLSCs’ caused effects on CD4^+^ T lymphocyte proliferation and apoptosis at different conditions. After 5 days incubation, CD4^+^ T lymphocyte proliferation and apoptosis were assessed by flow cytometry analysis.

#### 2.4.3. Immunomediators Expression

Primary hPDLSCs were seeded at 6-well plates in 3 ml DMEM supplemented with 10% FBS and 1% P/S at a density of 2.5 × 10^5^ cells per well. After 24 h, hPDLSCs were treated as indicated above. Then, 48 h later, IDO-1, PD-L1, PD-L2, and PTGS-2 gene and protein expression levels were measured. Additionally, IDO-1 enzymatic activity was measured.

### 2.5. Analysis of Co-Culture Experiments

#### 2.5.1. CD4^+^ T Cell Proliferation and Apoptosis

Isolated CD4^+^ T lymphocytes were labeled using CellTrace CFSE Cell Proliferation Kit (ThermoFischer Scientific, Waltham, USA) to measure CD4^+^ T lymphocyte proliferation. Briefly, isolated CD4^+^ T lymphocytes were resuspended in pre-warmed 5% FBS containing 1xPBS, getting a final CD4^+^ T lymphocyte concentration of 1 × 10^6^ cells per milliliter. CD4^+^ T lymphocytes were labeled with 2.5 µM carboxyfluorescein succinimidyl ester (CFSE) for 5 minutes at room temperature. Labeled CD4^+^ T lymphocytes were resuspended in complete RPMI-1640 and applied to indirect co-culture. After 5 days, CD4^+^ T lymphocytes were harvested and stained for apoptotic cells using 20 µg/ml propidium iodide (Pi, Affymetrix, Santa Clara, USA). CD4^+^ T lymphocytes were analyzed by flow cytometry using FACSCalibur Flow Cytometer (Becton Dickinson, Franklin Lakes, USA) equipped with an argon laser to excite fluorescence at 488 nm. Cell count was limited to 10,000 cells per sample, and analysis was performed using CellQuest 3.3. software (Becton Dickinson, Franklin Lakes, USA). The percentage of at least once divided CD4^+^ T lymphocytes was calculated. Additionally, the percentage of apoptotic cells was determined by measuring Pi positive cells. Representative dot plots that outline the gating/analysis strategy are shown in the Appendix A.

#### 2.5.2. Immunomediators Expression Analysis in hPDLSCs

##### Quantitative Polymerase Chain Reaction (qPCR)

Cell lysis, mRNA extraction, reverse transcription, and qPCR were performed using TaqMan Gene Expression Cells-to-CT kit (Applied Biosystems, Foster City, USA). Primus 96 advanced thermocycler (PeqLab/VWR, Darmstadt, Germany) was used for reverse transcription, heating the samples to 37 °C for 1 h followed by 95 °C for 5 minutes. qPCR was performed on a QuantStudio 3 device (Applied Biosystems, Foster City, USA) using the following settings: 1 x 95 ° C for 10 minutes followed by 50 cycles of 15 seconds at 95 °C and 1 minute at 60 °C. The following Taqman Gene Expression Assays (Applied Biosystems, Foster City, USA) were used for target genes amplification: IDO, Hs00984148_m1; PTGS-2, Hs00153133_m1; PD-L1, Hs00204257_m1; PD-L2, Hs00228839_m1 and GAPDH, Hs99999905. GAPDH served as internal reference. Target gene amplification was performed in paired reactions followed by determining C_t_ values. The n-fold gene expression of the target genes compared to untreated control was quantified by the 2^−∆∆Ct^ method.

##### IDO-1 Immunostaining

The 2.5 × 10^5^ hPDLSCs were fixed and permeabilized using the Intracellular Fixation and Permeabilization Buffer Set (ebioscience, Waltham, USA) and were subsequently stained with PE-conjugated mouse anti-human IDO-1 antibody (clone eyedio, ebioscience, Waltham, USA). Cells stained with PE-conjugated mouse IgG1 K immunoglobulin isotype control (ebioscience, Waltham, USA) served as reference. After staining, hPDLSCs were resuspended in 200 µl FACS buffer (3% bovine serum albumin and 0.09% sodium azid in 1xPBS) and analyzed by flow cytometry using the FACSCalibur Flow Cytometer. Fluorescence was excited by an argon laser at 488 nm. In total, 10,000 cells were counted per group. The percentage of IDO-1 positive cells and the corresponding mean fluorescence intensity (m.f.i.) were determined using CellQuest 3.3. software as described previously [24]. Representative dot plots and one-parameter histograms that demonstrate the gating/analysis strategy are provided in the Appendix A.

##### Measuring L-Kynurenine Levels

L-kynurenine levels were determined in conditioned media and in cell lysates. For cell lysates, conditioned media were harvested, and hPDLSCs were incubated for 3 h in 0.5 ml 1xPBS supplemented with 800 µM L-tryptophan (Sigma-Aldrich, St. Louis, USA). Conditioned media and cell lysates were mixed 1:3 (v/v) with 30% trichloroacetic acid (Sigma-Aldrich, St. Louis, USA) and were incubated at 65 °C for 30 minutes. 125 µl supernatant was mixed 1:1 with Ehrlich’s Reagent (0.8% P-dimethylbenzaldehyde in glacial acetic acid, Sigma-Aldrich, St. Louis, USA). After incubation for 10 minutes at room temperature, samples were measured at OD_492_ in duplicates. L-kynurenine concentrations were determined by plotting measured OD_492_ against known L-kynurenine (Sigma-Aldrich, St. Louis, USA) concentrations ranging from 1000 µM to 7.8 µM.

Total protein amounts were determined using Pierce BCA Protein Assay Kit (ThermoFischer Scientific, Waltham, USA). Protein amounts were determined by plotting measured OD_562_ against bovine serum album standards (GE Healthcare, Chicago, USA) ranging from 2000 µg/ml to 31.25 µg/ml. Determined L-kynurenine concentrations were normalized to total protein amounts.

##### PD-L1 and PD-L2 Immunostaining

1.25 × 10^5^ harvested hPDLSCs were resuspended in 50 µl FACS buffer and stained with the following antibodies for 20 minutes: PE-conjugated mouse anti-human CD274 antibody (clone B7-H1, ebioscience, Waltham, USA) or PE-conjugated mouse anti-human CD273 antibody (clone B7-DC, ebioscience, Waltham, USA). Unlabeled cells stained with (PE)-conjugated mouse IgG1 K immunoglobulin isotype control (ebioscience, Waltham, USA) served as control. After staining, cells were resuspended in 200 µl FACS buffer. Flow cytometry analysis was performed as described above. The percentage of PD-L1 or PD-L2 positive cells was determined using CellQuest 3.3 software. Representative dot plots that outline the used gating/analysis strategy are shown in the Appendix A.

##### Prostaglandin E_2_ ELISA

Prostaglandin E_2_ levels were detected in the appropriate conditioned media using Prostaglandin E2 Parameter Assay Kit (R&D Systems, Minneapolis, USA).

### 2.6. Statistical Analysis

All statistical analysis was performed using SPSS 24.0 (IBM, Armonk, USA). Calculating the statistical significances between groups was performed by Friedmann test followed by Wilcoxon test for pairwise comparison. All data are presented as mean values ± S.E.M (standard error of the mean). *p*-values < 0.05 were considered to be statistically significant.

## 3. Results

### 3.1. Mesenchymal and Hematopoietic Surface Marker Expression in hPDLSCs

According to the International Society for Cell and Gene Therapy [2,25], the stem cell character of hPDLSCs was verified by determining the expression of typical mesenchymal and hematopoietic stem cell surface markers (Table 1). hPDLSCs were stained positively (> 95%) for MSC surface markers CD29, CD73, CD90, and CD105. Approximately 61% of hPDLSCs were stained positive for MSC surface marker CD146. Additionally, hPDLSCs were stained negatively (< 3%) for hematopoietic surface markers CD31, CD34 and CD45.

### 3.2. hPDLSC Mediated Effect of Different Inflammatory Stimuli on CD4^+^ T Lymphocytes

Figure 1a shows the effect of different inflammatory cytokines on PHA-induced CD4^+^ T lymphocyte proliferation in the absence and the presence of hPDLSCs. In the absence of hPDLSCs, CD4^+^ T lymphocyte proliferation was significantly inhibited by TNF-α and not affected by IL-1β and IFN-γ. hPDLSCs significantly inhibited CD4^+^ T lymphocyte proliferation, even in the absence of cytokines. The inhibition of CD4^+^ T lymphocyte proliferation by hPDLSCs was significantly enhanced by IL-1β and IFN-γ, while it was not affected by TNF-α. Figure 1b shows the effect of different inflammatory cytokines on the apoptosis of PHA-activated CD4^+^ T lymphocytes in the presence of hPDLSCs. In the absence of any cytokine, hPDLSCs significantly decreased the PHA-induced apoptosis of CD4^+^ T lymphocytes. However, IL-1β and IFN-γ treatment of hPDLSCs counteracted the reduction of CD4^+^ T lymphocyte apoptosis, causing a significant increase in the percentage of Pi positive CD4^+^ T lymphocytes to a similar extent. Both cytokines resulted in a significantly higher apoptosis than TNF-α stimulated hPDLSCs, which did not affect the percentage of apoptotic CD4^+^ T lymphocytes.

### 3.3. Effect of Different Inflammatory Cytokines on the Expression of Immunomodulatory Proteins in hPDLSCs

#### 3.3.1. IDO-1 Expression and Activity

Figure 2 shows the effects of IL-1β, TNF-α, and IFN-γ on the IDO-1 gene and protein expression levels as well as on the IDO-1 enzymatic activity. Each cytokine induced significantly enhanced IDO-1 gene expression levels compared to the control (Figure 2a). Significant differences were found between different treatments; the highest IDO-1 gene expression level was observed after IFN-γ treatment followed by TNF-α and IL-1β treatment. All inflammatory stimuli significantly increased the percentage of IDO-1 positive hPDLSCs (Figure 2b) and the corresponding m.f.i. (Figure 2c). Significant differences were observed between different cytokines; the highest IDO-1 protein level was induced by IFN-γ followed by TNF-α and IL-1β. IDO-1 enzymatic activity was determined by measuring L-kynurenine concentrations in cell lysates (Figure 2d) and conditioned media (Figure 2e). In cell lysates, only IFN-γ was able to induce a significant increase in L-kynurenine production, whereas TNF-α and IL-1β had no significant effect. All three stimuli induced significantly higher L-kynurenine production in conditioned media; the highest effect was observed for IFN-γ followed by TNF-α and IL-1β.

#### 3.3.2. PD-L1/2 Expression

Figure 3 shows the effects of IL-1β, TNF-α, and IFN-γ on PD-L1 and PD-L2 gene and protein expression levels in hPDLSCs. IL-1β, TNF-α, and IFN-γ significantly enhanced PD-L1 gene expression in hPDLSCs (Figure 3a) and the proportion of PD-L1 positive cells (Figure 3c). On gene level, the effect of IFN-γ was significantly higher than those of TNF-α and IL-1β. The proportion of PD-L1 positive cells after treatment with IFN-γ or TNF-α was significantly higher compared to IL-1β treatment. The gene expression level of PD-L2 (Figure 3b) was significantly higher upon the stimulation with IFN-γ or TNF-α by similar extent. Both, IFN-γ and TNF-α induced a significant increase in the proportion of PD-L2 positive cells (Figure 3d), but the effect of IFN-γ was significantly higher than that of TNF-α. No significant effect of IL-1β on PD-L2 gene and protein expression was observed.

#### 3.3.3. PTGS-2/PGE_2_ Expression

Figure 4 shows the effects of IL-1β, TNF-α, and IFN-γ on PTGS-2/PGE_2_ expression in hPDLSCs. All investigated inflammatory factors induced a significantly enhanced PTGS-2 gene expression (Figure 4a) and PGE_2_ production (Figure 4b). The effect of IL-1β was significantly higher than those of IFN-γ and TNF-α. No significant differences between the effects of TNF-α and IFN-γ on PTGS-2 and PGE_2_ production were detected.

### 3.4. Effect of IDO-1, PD-L1, and PTGS-2 Inhibition in hPDLSCs on CD4^+^ T Lymphocyte Proliferation in the Presence of Different Inflammatory Stimuli

Figure 5 shows how the inhibition of IDO-1 (Figure 5a), PD-L1 (Figure 5b), and PGE_2_ (Figure 5c) influences the effect of hPDLSCs on PHA-induced CD4^+^ T lymphocyte proliferation in the presence or the absence of IFN-γ, TNF-α, or IL-1β. In the presence of different cytokines, pharmacological inhibition of IDO-1, PD-L1, or PTGS-2 counteracted hPDLSC-induced suppression of CD4^+^ T lymphocytes to different degrees. IDO-1 inhibition (Figure 5a) in hPDLSCs resulted in a significant increase of CD4^+^ T lymphocyte proliferation in the presence of IFN-γ and its decrease in the presence of IL-1β. No significant effect of IDO-1 inhibition on CD4^+^ T lymphocyte proliferation was observed in the presence of TNF-α. PD-1/PD-L1 inhibition (Figure 5b) significantly increased CD4^+^ T lymphocyte proliferation only in the presence of IFN-γ treated hPDLSCs. PTGS-2 inhibition (Figure 5c) significantly increased CD4^+^ T lymphocyte proliferation in the presence of hPDLSCs and the absence of any cytokine. Similar effects were also observed in the presence of all cytokines, but its extent differed depending on the cytokine type; the most prominent effect of PTGS-2 inhibition was observed in the presence of IL-1β.

### 3.5. Effect of IDO-1, PD-L1, and PTGS-2 Inhibition in hPDLSCs on CD4^+^ T Lymphocyte Apoptosis in the Presence of Different Inflammatory Stimuli

Figure 6 shows how IDO-1 (Figure 6a), PD-L1 (Figure 6b), or PGE_2_ (Figure 6c) inhibition in hPDLSCs affect apoptosis of PHA-activated CD4^+^ T lymphocytes in the presence different cytokines. In the presence of hPDLSCs and the absence of any cytokine, the inhibition of IDO-1, PD-L1, or PTGS-2 had no effect on CD4^+^ T lymphocyte apoptosis. In the presence of different cytokines, pharmacological inhibition of IDO-1, PD-L1, or PTGS-2 in hPDLSCs differently affected CD4^+^ T lymphocyte apoptosis. IDO-1 inhibition (Figure 6a) showed a significant increase in apoptotic CD4^+^ T lymphocytes only in the presence of IFN-γ treated hPDLSCs. PD-1/PD-L1 (Figure 6b) inhibition had no significant effect on CD4^+^ T lymphocyte apoptosis. PTGS-2 (Figure 6c) inhibition caused a significant decrease in Pi positive CD4^+^ T lymphocytes in the presence of IL-1β treated hPDLSCs. In the presence of TNF-α triggered hPDLSCs, none of the tested inhibitors affected the percentage of apoptotic CD4^+^ T lymphocytes.

## 4. Discussion

Intensive preclinical research in the last decade declared MSC-like cells to be a promising therapeutic candidate in the fields of regenerative medicine [11] and inflammatory diseases [26]. Numerous clinical trials using MSC-based therapeutic approaches [27] exhibited only moderate to no success. Nowadays, the therapeutic potential of transplanted MSCs is mainly associated with their immunomodulatory and secretory activities [28]. This assumption is reinforced by the recent observation that the efficacy of MSC-based therapies is enhanced through stimulating their immunomodulatory potential by ex vivo treatment with IFN-γ [29]. Therefore, the regulation of the immunomodulatory ability of hPDLSCs by different cytokines needs to be understood in detail.

The concentrations of the cytokines used were chosen based on both clinical relevance and preliminary data. We chose the percentage of IDO-1 positive hPDLSCs as a target parameter for our preliminary experiments, because this protein is assumed to be the main mediator of MSCs’ immunomodulatory function. The 5 ng/ml IL-1β and the 100 ng/ml IFN-γ reflect the corresponding cytokine levels in the gingival crevicular fluid (GCF) of periodontitis patients [30,31]. Additionally, the chosen IL-1β and IFN-γ levels caused the highest percentage of IDO-1 positive hPDLSCs within our experimental in vitro setting (Appendix A). The used TNF-α concentration (10 ng/ml) was clearly higher than in the GCF of periodontitis patients (up to 100 pg/ml) [32]. The percentage of IDO-1 positive hPDLSCs caused by 10 ng/ml TNF-α was near the maximal response (90% of the response caused by 100 ng/ml TNF-α). Furthermore, most previous studies used TNF-α at a concentration of 10 ng/ml [20].

Our data showed that the effect of hPDLSCs on the proliferation of CD4^+^ T lymphocytes is differently affected by various inflammatory cytokines. hPDLSCs inhibited the proliferation of CD4^+^ T lymphocytes, even in the absence of any cytokine. This inhibitory effect was strongly enhanced by 100 ng/ml IFN-γ and 5 ng/ml IL-1β. The additional inhibitory effect is mediated by hPDLSCs because these cytokines had no significant effect on CD4^+^ T lymphocyte proliferation in the absence of hPDLSCs. In contrast, 10 ng/ml TNF-α showed no potentiating effect of the hPDLSCs mediated suppression of CD4+ T cell proliferation. This was in contrast to the direct inhibitory effect of 10 ng/ml TNF-α on CD4^+^ T lymphocyte proliferation observed in the absence of hPDLSCs. Thus, the presence of hPDLSCs modifies the effect of all investigated cytokines on CD4^+^ T lymphocyte proliferation.

The changes of immunomodulatory properties of MSCs on CD4^+^ T lymphocyte proliferation by inflammatory cytokines was already partially indicated in previous studies. The inhibitory effects of MSCs on CD4^+^ T lymphocyte proliferation has only been reported in simultaneous presence of IFN-γ with TNF-α or IL-1β using 20 ng/ml for each cytokine [33]. Using bone marrow-derived MSCs from mice and antibody-activated CD4^+^ T lymphocyte may be the major reasons for these partial discrepancies in our data. Additionally, using a fivefold lower IFN-γ concentrations may also be the reason for the unaffected T lymphocyte proliferation by IFN-γ treated MSCs. Another study reports no significant effect of 1 U/ml IFN-γ and TNF-α on peripheral blood-derived human MSCs mediated suppression of antibody-triggered CD4^+^ T lymphocyte proliferation [34]. It seems as if suppressive effects of such cytokines depend on several factors, such as MSC origin or CD4^+^ T lymphocyte activation method.

Our results further indicated that the effect of hPDLSCs on PHA-triggered apoptosis of CD4^+^ T lymphocytes is differently influenced by various inflammatory cytokines. In the absence of any cytokine, hPDLSCs per se reduced the number of apoptotic CD4^+^ T lymphocytes. Consequently, this finding implies that hPDLSCs protect CD4^+^ T lymphocytes from PHA-induced apoptosis. Furthermore, it seems that hPDLSC-mediated inhibition of CD4^+^ T lymphocyte proliferation is not due to CD4^+^ T lymphocyte apoptosis. This is in accordance with previous studies that observed that bone marrow derived MSC-mediated suppression of T lymphocytes does not depend on cellular apoptosis [35] but rather on arresting T lymphocytes in the G0/G1 cell cycle phase [36]. In contrast, other studies conducted in dental pulp stem cells [37] and hPDLSCs [16] showed an increase in the apoptosis of T lymphocytes in the absence of any cytokine. Using dental pulp stem cells [37] and concanavalin A [16] stimulated PBMCs, respectively, may be the major reasons for the discrepancies in our data. 

5 ng/ml IL-1β or 100 ng/ml IFN-γ partially counteracted the hPDLSCs’ induced reduction of apoptotic CD4^+^ T lymphocytes, whereas 10 ng/ml TNF-α slightly increased the anti-apoptotic effect of hPDLSCs on CD4^+^ T lymphocytes. Thus, adding various exogenous cytokines to our co-culture model partially caused the usual known immunosuppressive properties of hPDLSCs and significantly modified the hPDLSCs’ ability to influence apoptosis of CD4^+^ T lymphocytes. Increased CD4^+^ T lymphocytes apoptosis by 100 ng/ml IFN-γ or 5 ng/ml IL-1β might partially underline the enhanced anti-proliferative effect of hPDLSCs on CD4^+^ T lymphocytes in the presence of these cytokines. Although 10 ng/ml TNF-α showed no effect on CD4^+^ T lymphocyte proliferation in the presence of hPDLSCs, this cytokine caused a decrease in the proportion of apoptotic CD4^+^ T lymphocytes. This anti-apoptotic effect of 10 ng/ml TNF-α treated hPDLSCs might partially explain the observation that 10 ng/ml TNF-α caused an inhibition of CD4^+^ T lymphocyte proliferation in the absence of hPDLSCs but did not affect it in the presence of hPDLSCs; the anti-proliferative effect of TNF-α might be counterweighed by its anti-apoptotic effect. Summarizing, it seems that, in the presence of cytokine treated hPDLSCs, apoptosis and proliferation of CD4^+^ T lymphocytes are highly connected to each other, which also may depend on the present cytokine type. This raises the question if cytokine treatment of hPDLSCs changes their way to suppress CD4^+^ T lymphocyte proliferation from an apoptosis independent to a partially apoptosis dependent mechanism. 

The most important factors mediating hPDLSCs immunomodulatory activities toward T lymphocytes are IDO-1, PD-L1, PD-L2, and PTGS-2 [5]. The expression of most of these immunomediators in hPDLSCs was increased by stimulation with 100 ng/ml IFN-γ, 10 ng/ml TNF-α, and 5 ng/ml IL-1β. However, prominent quantitative differences were observed between different cytokines. Few previous studies compare the effect of different cytokines in hPDLSCs, and their results are partly in line with our data. Most previous studies suggest that IDO-1 protein expression and activity could be enhanced by all cytokines, such as 0.5–50 ng/ml IL-1β and TNF-α but could be dramatically increased only by IFN-γ at concentrations of 10–25 ng/ml [16,17,18]. Although partially using different concentrations and incubation times, ranging from 24 h to 72 h, these studies are in line with our data. Another study shows that 10 ng/ml TNF-α induces significantly higher PD-L1 surface expression in hPDLSCs compared to 10 ng/ml IFN-γ or IL-1β after 48 h [20]. In contrast, in our study, 10 ng/ml TNF-α and 100 ng/ml IFN-γ caused quantitatively similar high PD-L1 protein expression levels, whereas 5 ng/ml IL-1β induced PD-L1 expression was rather low. Further studies imply that TNF-α (1 – 10 ng/ml) is a strong activator of PGE2 production in hPDLSCs [38]. Another study showed similar PGE2 levels after stimulating MSCs with 100 ng/ml IFN-γ or 10 ng/ml TNF-α for 48 h [21], which is comparable to our data. However, this study further reported that the production of PGE-2 by MSCs was drastically increased by 50 ng/ml TNF-α. We found the strongest production of PGE-2 protein after stimulation with IL-1β. The expression level of PD-L2 was also different after stimulation with various cytokines, which was not shown previously.

The mechanisms involved in the immunomodulatory activity of hPDLSCs after activation by different cytokines are complex and seem to depend on the inflammatory environment. Our results identified PGE_2_ as a potent immunomediator of CD4^+^ T lymphocyte proliferation. Pharmacological inhibition of PTGS-2 counteracted the suppression of CD4^+^ T lymphocyte proliferation in the presence of all three investigated cytokines. PGE_2_ mediated mechanisms almost fully reversed the hPDLSCs mediated suppression of CD4^+^ T lymphocytes proliferation in the presence of 5 ng/ml IL-1β. This finding is in agreement with observation that 5 ng/ml IL-1β induced the highest PGE_2_ expression levels in hPDLSCs compared to other investigated cytokines. Additionally, PTGS-2 inhibitor reversed hPDLSC mediated effects on CD4^+^ T lymphocyte proliferation, even in the absence of any cytokine. This observation could be explained by the assumption of the reciprocal interaction between hPDLSCs and CD4^+^ T lymphocytes in our in vitro co-culture model. Particularly, cytokines produced by activated CD4^+^ T lymphocytes might activate PGE_2_ production by hPDLSCs, which, in turn, might suppress CD4^+^ T lymphocyte proliferation. A previous study implies that PGE_2_ production is an important mechanism involved in the immunomodulatory effects of MSCs mainly toward macrophages [5]. Our study additionally underlined the importance of this factor in MSCs-mediated effects on CD4^+^ T lymphocytes.

Previous reports suggest that MSC-mediated suppression of CD4^+^ T lymphocyte proliferation is mainly regulated by the IFN-γ–IDO-1 axis [4,39,40]. This statement is only partially in line with our data. On the one hand, the suppression of CD4+ T cell proliferation by hPDLSCs was strongly promoted by 100 ng/ml IFN-γ. Moreover, gene and protein expression of IDO-1 in hPDLSCs was mostly enhanced by 100 ng/ml IFN-γ stimulation compared to other cytokines. On the other hand, pharmacological inhibition of IFN-γ-induced IDO-1 expression only partially counteracted the suppression of CD4^+^ T lymphocytes by hPDLSCs. Furthermore, IDO- inhibitor had no effect on the hPDLSCs-mediated inhibition of CD4^+^ T lymphocytes proliferation in the presence of 10 ng/ml TNF-α and 5 ng/ml IL-1β as well as in the absence of any cytokines. Moreover, 10 ng/ml TNF-α induced high levels of IDO-1 expression but failed to enhance the inhibitory effect of hPDLSCs on CD4^+^ T lymphocyte proliferation. Thus, although IDO-1 is an important immunomodulatory factor, it cannot account for all hPDLSCs-mediated effects on CD4^+^ T lymphocytes proliferation.

PD-L1 is an important transmembrane protein involved in suppressing immune responses. Previous studies also show its implication in MSC-mediated immunosuppressive effects on T lymphocytes [20]. Zhang et al. demonstrated that PD-L1 is a crucial factor in mediating the suppression of T lymphocytes by MSCs derived from the adipose tissue in the absence of any cytokine [41]. This is in contrast with our data, which show that only the inhibition of 100 ng/ml IFN-γ induced PD-L1 in hPDLSCs counteracted the suppression of CD4^+^ T lymphocytes. This is in accordance with our gene expression analysis data showing that 100 ng/ml IFN-γ caused the highest PD-L1 expression levels. Hence, it seems that 100 ng/ml IFN-γ induced PD-L1 is another important axis for suppressing CD4^+^ T lymphocyte proliferation by hPDLSCs. However, this mechanism seems to be less relevant in the presence of 10 ng/ml TNF-α and 5 ng/ml IL-1β, although its expression in hPDLSCs is substantially increased by these cytokines.

In contrast to CD4^+^ T lymphocytes proliferation, the effect of different pharmacological inhibitors on the hPDLSCs-mediated effect on CD4^+^ T lymphocytes apoptosis was less pronounced. None of the inhibitors influenced CD4^+^ T lymphocytes apoptosis in the presence of hPDLSCs and the absence of cytokines. The IDO-1 inhibitor marginally increased the proportion of apoptotic CD4^+^ T lymphocytes in the presence of 100 ng/ml IFN-γ. The PTGS-2 inhibitor markedly increased the proportion of apoptotic CD4^+^ T lymphocytes in the presence of 100 ng/ml IFN-γ, while it was decreased in the presence of 5 ng/ml IL-1β. The inhibition of PD-L1 caused no influence on the hPDLSCs-mediated CD4^+^ T lymphocyte apoptosis, although Zhang et al. showed a contribution of 10 ng/ml TNF-α induced PD-L1 expression in hPDLSCs on CD4^+^ T lymphocytes apoptosis. This discrepancy might be explained by the use of PBMCs in a direct co-culture model [20] instead of pure CD4^+^ T lymphocytes in our indirect co-culture model. Summarizing, it seems that the mechanisms of hPDLSCs-mediated effects on CD4+ T lymphocytes apoptosis depend on the inflammatory microenvironment conditions.

Our study underlies the complexity of the interaction between hPDLSCs and CD4^+^ T lymphocytes and further indicates a high plasticity of the reciprocal interaction between hPDLSCs and immune cells mainly regulated by the local cytokine milieu. IFN-γ, TNF-α, and IL-1β are mainly secreted by various activated immune cells under inflammatory conditions. These cytokines activate resting hPDLSCs and trigger the expression of various immunomediators in hPDLSCs and consequently their immunosuppressive activities. This causes lower levels of inflammatory cytokines and subsequently a partial abolishment of hPDLSCs’ immunosuppression. Thus, a continuous interaction between hPDLSCs and immune cells seems to determine the intensity of the local immune response (reviewed in [5]). Our results indicate that inflammatory cytokines differently affect the production of immunomodulatory factors in hPDLSCs and consequently their immunomodulatory activities. It can be hypothesized that a certain inflammatory cytokine predominantly activates only a particular immunomodulatory mechanism and consequently only a specific immunomodulatory function. This could lead to variable immunosuppressive actions on immune cells and subsequently to variable cytokine levels. Hence, the plasticity of this tight bidirectional interaction is not only affected by the cytokine level [28] but also by the kind of cytokine. Both parameters significantly change during the inflammation process. With regard to the variable cytokine levels during the inflammation progress, one limitation of our study is the use of only one concentration per cytokine.

The complex interaction between hPDLSCs and CD4^+^ T lymphocytes might have an implication in pathological states, particularly in periodontal disease. Periodontitis is an inflammatory disease of the tooth supporting tissues that might lead to tooth loss if it remains untreated. T lymphocytes play a crucial role in periodontitis progression by contributing to osteoclast formation and alveolar bone resorption and by contributing to a pro-inflammatory environment [42]. It is also known that tissue destruction in periodontal disease is mainly caused by a dysregulated immune response. All used cytokines are involved in a host destructive inflammatory response and are usually associated with alveolar bone resorption by triggering osteoclastogenesis and soft tissue degradation by matrix metalloproteinases [43,44,45]. Further, our data implies that, besides destructive effects, these cytokines might also activate some immunosuppressive mechanisms. Modulation of the activity of immune cells by hPDLSCs and influencing this effect differently by inflammatory cytokines represents a fine-tuning mechanism of the local inflamed periodontal microenvironment.

## 5. Conclusions

In conclusion, this in vitro study underlines that the immunomodulatory activity of hPDLSCs strongly depends on the local microenvironment. Depending on the presence of different inflammatory cytokines, the expression of several immunomediators is upregulated, which causes variable effects on CD4^+^ T cells. By translating our results into the in vivo situation, we suggest that the regulation of local immune responses by hPDLSCs in the presence of various cytokines is a crucial mechanism of on-site immune and periodontal tissue homeostasis.

## Figures and Tables

**Figure 1 cells-09-01222-f001:**
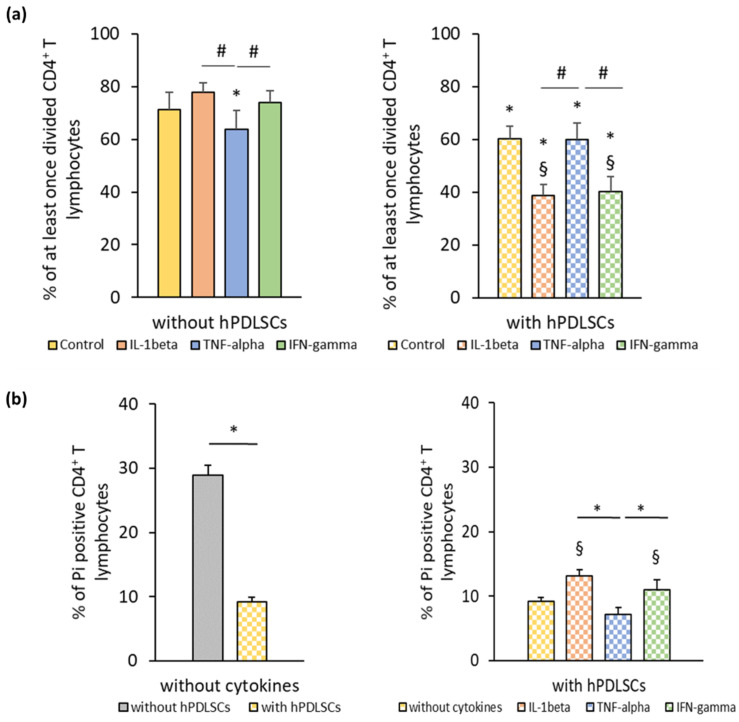
hPDLSC mediated effect of different inflammatory stimuli (IL-1β, TNF-α, and IFN-γ) on CD4^+^ T lymphocyte proliferation and apoptosis. Primary hPDLSCs were stimulated with either 5 ng/ml IL-1β or 10 ng/ml TNF-α or 100 ng/ml IFN-γ. After 48 h, hPDLSCs were applied to indirect co-culture and stimulated as described above. Allogenic CD4^+^ T lymphocytes were added to indirect co-culture using Transwell (TC) inserts. Additionally, CD4^+^ T lymphocytes were cultured in the absence of hPDLSCs and stimulated with either 5 ng/ml IL-1β or 10 ng/ml TNF-α or 100 ng/ml IFN-γ. CD4^+^ T lymphocyte proliferation was induced by 10 µg/ml phytohemagglutinin-L (PHA-L). After five days incubation, CD4^+^ T lymphocyte proliferation (**a**) was determined by flow cytometry by analyzing carboxyfluorescein succinimidyl ester (CFSE) labeled CD4^+^ T lymphocytes. The Y-axis shows the percentage of at least once divided CD4^+^ T lymphocytes. Additionally, after five days incubation, the percentage of apoptotic CD4^+^ T lymphocytes (**b**) was determined by flow cytometry by labeling CD4^+^ T lymphocytes with Pi. The Y-axis shows the percentage of Pi positive CD4^+^ T lymphocytes. Data are presented as mean value ± S.E.M. received from five independent experiments with hPDLSCs from five different individuals. * *p*-value < 0.05 compared to the control (without hPDLSCs); § *p*-value < 0.05 compared to CD4^+^ T lymphocytes in the presence of untriggered hPDLSCs; # *p*-value < 0.05 compared between groups as indicated.

**Figure 2 cells-09-01222-f002:**
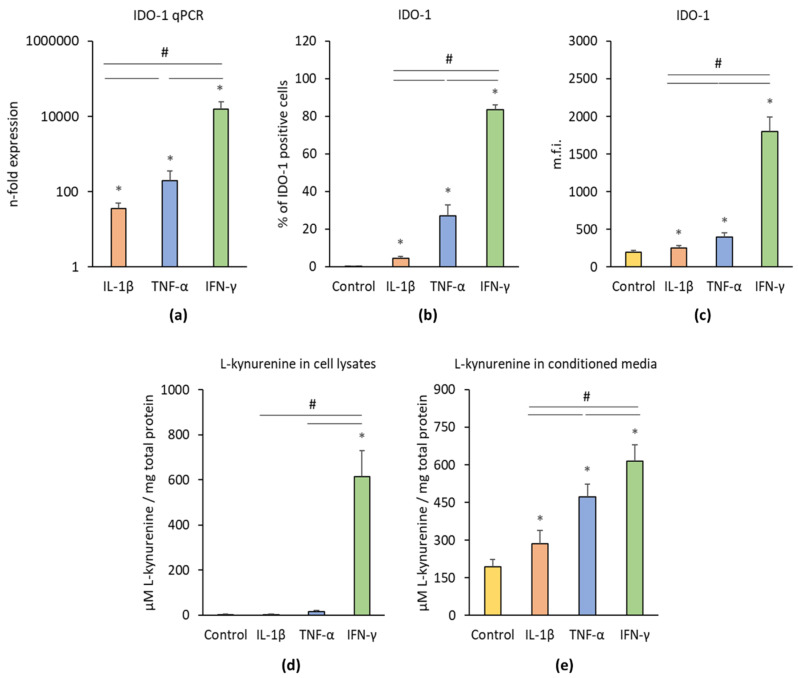
Effect of pro-inflammatory stimuli (IL-1β, TNF-α, and IFN-γ) on intoleamine-2,3-dioxygenase-1 (IDO-1) production and activity in hPDLSCs. Primary hPDLSCs were treated with either 5 ng/ml IL-1β or 10 ng/ml TNF-α or 100 ng/ml IFN-γ. Unstimulated hPDLSCs served as control. After 48 h treatment, IDO-1 gene expression levels (**a**) were measured using qPCR, demonstrating the n-fold IDO-1 expression compared to the control (n = 1). GAPDH served as internal reference gene. Corresponding IDO-1 protein levels were quantified by intracellular immunostaining followed by flow cytometry analysis, determining the percentage of IDO-1 positive hPDLSCs (**b**) and the corresponding mean fluorescence intensity (m.f.i.) (**c**). IDO-1 enzymatic activity was measured by quantifying L-kynurenine concentration (µM) in cell lysates (**d**) and the conditioned media (**e**) normalized to the total protein amounts in mg. All data are presented as mean value ± S.E.M. received from six independent experiments with cells isolated from six different individuals. * *p*-value < 0.05 compared to the unstimulated control; # *p*-value < 0.05 compared to appropriate groups as indicated.

**Figure 3 cells-09-01222-f003:**
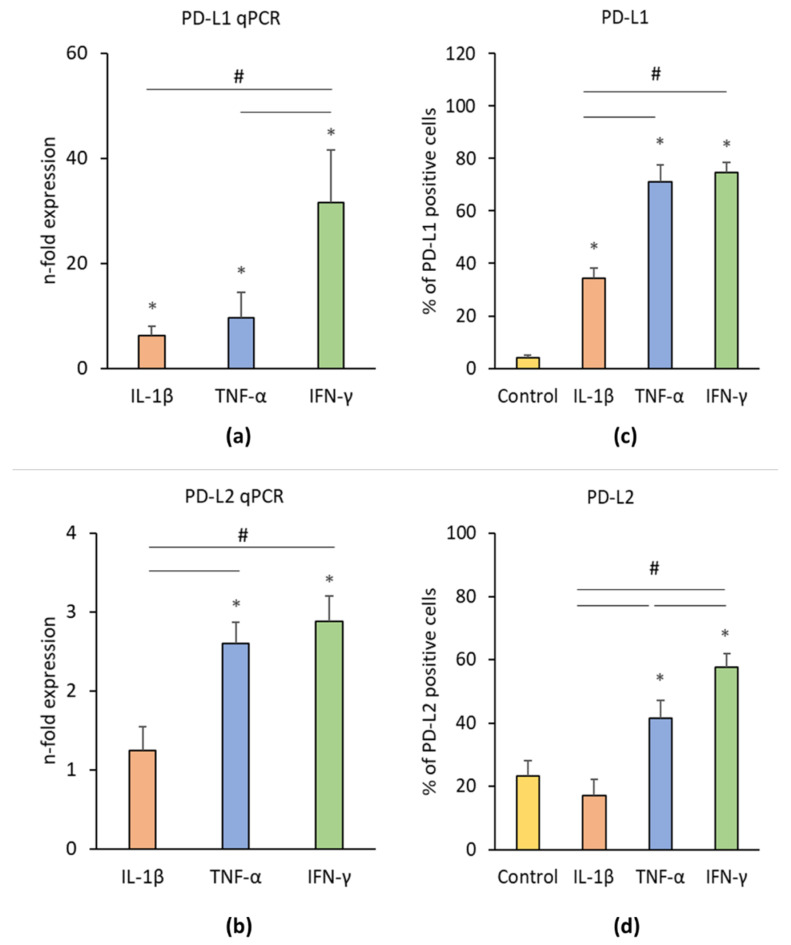
Effect of pro-inflammatory stimuli (IL-1β, TNF-α, and IFN-γ) on programmed cell death 1 ligand 1 (PD-L1) and programmed cell death 1 ligand 2 (PD-L2) production in hPDLSCs. Primary hPDLSCs were treated with either 5 ng/ml IL-1β or 10 ng/ml TNF-α or 100 ng/ml IFN-γ for 48 h. Unstimulated hPDLSCs served as control. PD-L1 (**a**) and PD-L2 (**b**) gene expression levels were determined by qPCR, demonstrating the n-fold PD-L1 / PD-L2 expression compared to the control (n = 1). GAPDH served as internal reference gene. PD-L1/PD-L2 protein levels were investigated by intracellular immunostaining followed by flow cytometry analysis, determining the percentage of PD-L1 (**c**) and PD-L2 (**d**) positive cells. All data are presented as mean value ± S.E.M. received from six independent experiments with cells isolated from six different individuals. * *p*-value < 0.05 compared to the unstimulated control; # *p*-value < 0.05 compared between groups as indicated.

**Figure 4 cells-09-01222-f004:**
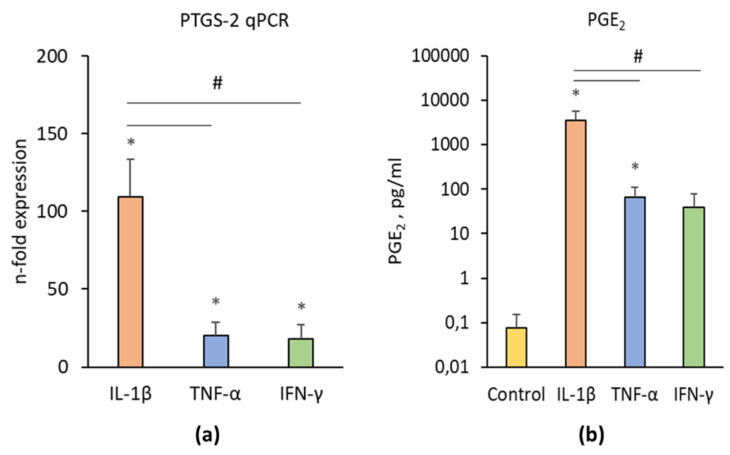
Effect of pro-inflammatory stimuli on PTGS-2 and prostaglandin E_2_ (PGE_2_) production in hPDLSCs. Primary hPDLSCs were treated with either 5 ng/ml IL-1β or 10 ng/ml TNF-α or 100 ng/ml IFN-γ for 48 h. Unstimulated hPDLSCs served as control. PTGS-2 gene expression (**a**) was determined by qPCR, demonstrating the n-fold PTGS-2 expression compared to the control (n-fold expression = 1). GAPDH served as internal reference gene. PGE_2_ production (**b**) was determined in conditioned media by a parameter assay. All data are presented as mean value ± S.E.M. received from six independent experiments with cells isolated from six different individuals. * *p*-value < 0.05 compared to the unstimulated control; # *p*-value < 0.05 compared to appropriate groups as indicated.

**Figure 5 cells-09-01222-f005:**
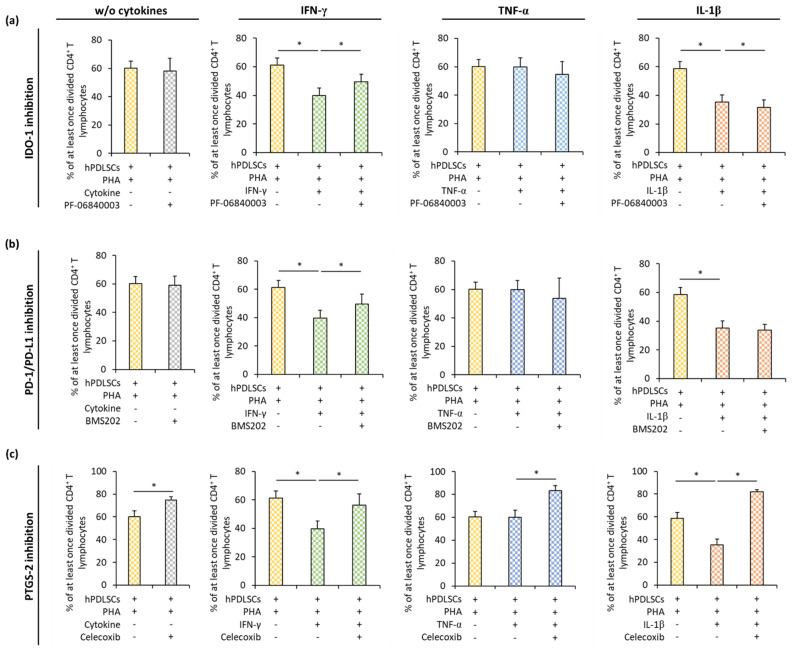
Effect of IDO-1, PD-L1, and PTGS-2 inhibition in hPDLSCs on CD4^+^ T lymphocyte proliferation in the presence of different inflammatory stimuli (IL-1β, TNF-α, and IFN-γ). Primary hPDLSCs were stimulated with either (**a**) 50 µM PF-06840003 (IDO-1 inhibitor) or (**b**) 1 µM BMS202 (PD-1/PD-L1 inhibitor) or (**c**) 1 µM Celecoxib (PTGS-2 inhibitor) in the absence or the presence of either 5 ng/ml IL-1β or 10 ng/ml TNF-α or 100 ng/ml IFN-γ. After 48 h, hPDLSCs were applied to indirect co-culture and stimulated as described above. Allogenic CD4^+^ T lymphocytes were added to indirect co-culture using TC inserts. CD4^+^ T lymphocyte proliferation was induced by 10 µg/ml PHA-L. After five days incubation, CD4^+^ T lymphocyte proliferation was determined by analyzing CFSE labeled CD4^+^ T lymphocytes via flow cytometry. The Y-axis shows the percentage of at least once divided CD4^+^ T lymphocytes. Data are presented as mean value ± S.E.M. originated from five independent experiments with hPDLSCs from five different individuals. * *p*-value < 0.05 compared between groups as indicated.

**Figure 6 cells-09-01222-f006:**
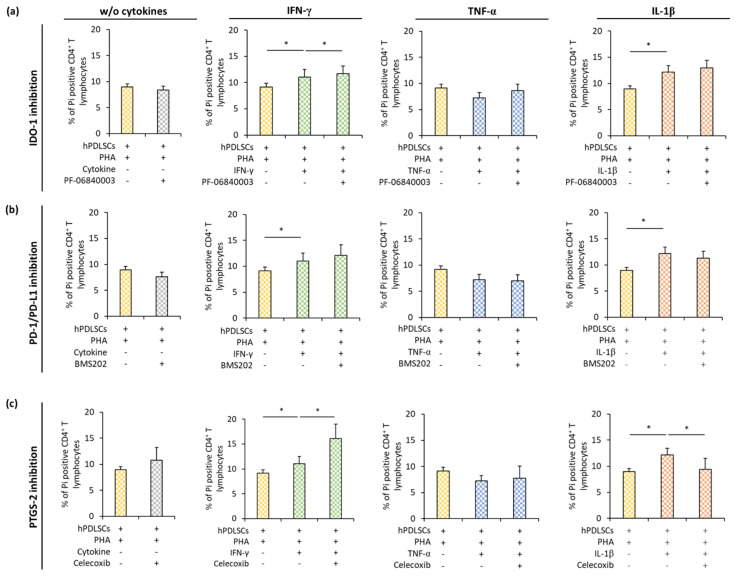
Effect of IDO-1, PD-L1, and PTGS-2 inhibition in hPDLSCs on CD4^+^ T lymphocyte apoptosis in the presence of different inflammatory stimuli (IL-1β, TNF-α, and IFN-γ). Primary hPDLSCs were stimulated with either (**a**) 50 µM PF-06840003 (IDO-1 inhibitor) or (**b**) 1 µM BMS202 (PD-1/PD-L1 inhibitor) or (**c**) 1 µM Celecoxib (PTGS-2 inhibitor) in the absence or the presence of either 5 ng/ml IL-1β or 10 ng/ml TNF-α or 100 ng/ml IFN-γ. After 48 h, hPDLSCs were applied to indirect co-culture and stimulated as described above. Allogenic CD4^+^ T lymphocytes were added to indirect co-culture using TC inserts. After five days incubation, the percentage of apoptotic CD4^+^ T lymphocytes was determined by flow cytometry by labeling CD4^+^ T lymphocytes with Pi. The Y-axis shows the percentage of Pi positive CD4^+^ T lymphocytes. Data are presented as mean value ± S.E.M. received from five independent experiments with hPDLSCs from five different individuals. * *p*-value < 0.05 compared between groups as indicated.

**Table 1 cells-09-01222-t001:** Human periodontal ligament stem cells’ (hPDLSCs) mesenchymal and hematopoietic surface marker expression analysis. Data are presented as mean values ± S.E.M. from five experiments using cells from five different donors. *MSC: human mesenchymal stem cells.

	MSC Marker	Hematopoietic Marker
**CD29**	97.7 ± 0.2	-
**CD73**	96.1 ± 0.2	-
**CD90**	97.9 ± 0.2	-
**CD105**	97.1 ± 0.6	-
**CD146**	61.3 ± 5.7	-
**CD31**	-	0.5 ± 0.1
**CD34**	-	0.6 ± 0.2
**CD45**	-	2.7 ± 0.2

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
