# Peer review of "Cytokines Differently Define the Immunomodulation of Mesenchymal Stem Cells from the Periodontal Ligament"

_cells, 2020, doi:10.3390/cells9051222_

Round 1
Reviewer 1 Report
The paper by Behm et al. is an interesting paper dealing with immunoregulatory effects of hPDLSCs. They evaluated some direct effects of IFN-γ, TNF-a and IL-1ß on hPDLSCs. Furthermore, they studied the effects of these cells on allogeneic CD4+ T lymphocyte proliferation and apoptosis.
The work is well written and the results are clearly presented. Minor typing errors are presents
The study is well designed and the rationale is clear. Although a lot has been studied and described about MSCs immunoregulation, the paper gives a contribution to the field, mainly because the results indicate how hPDLSCs could contribute to periodontal tissue homeostasis. Furthermore, the results confirm how complex is the crosstalk between MSCs and the immune system. To this aim, I would suggest broadening the discussion: the authors compare their results to similar paper point by point, but the comprehensive picture of the interrelationship of hPDLSCs and immune system (in particular pro-inflammatory bioactive molecules) could be more widely discussed.
I think the work could be published also in this form, although I think the discussion could be improved as suggested.
Author Response
Please, find our response in the attached file.

Reviewer 2 Report
[Cells] Manuscript ID: cells-757065
Cytokines differently define the immunomodulation of mesenchymal stem cells from the periodontal ligament
The paper studied effects of TNFa, IL-1b, and IFNg on the interaction of hPDLSCs with T-cells. The major parameters were PHA induced T-cell proliferation and apoptosis. On average all is done accurately. However the effects are rather marginal, most of the effects are studied earlier by different groups. The novelty is the comparison of three cytokines in one study.
Minor remarks
Different type of citation
“The most important inflammatory cytokines affecting MSCs are interferon (IFN)-γ, tumor necrosis factor (TNF)-α and interleukin (IL)-1β (Krampera 2011; El-Sayed et al. 2018).”
Fig 1 a – TNFa treated hPDLSCs stimulated T-cell proliferation?
Author Response

(The authors gave the same response as above.)

Reviewer 3 Report
Dear Authors, you here studied in methodic fashion how the immunomodulatory properties of periodontal ligament derived mesenchymal stromal cells (PDLSCs) are differentially affected by inflammatory licensing with different cytokines (IFN-gamma, TNF-alpha, IL1-beta). In resemblance to prior studies on MSCs derived from other tissues, you studied how major immunomodulatory signaling pathways and the expression of their associated typical effector molecules are regulated in response to these cytokines and how the inflammatory priming affects the immunomodulatory properties of PDLSCs on T-cell proliferation in vitro. To clearly distinguish yourself from the prior studies cited in references 16-18, you have emphasize the importance of your quantitative assessment for comparing PDLSC licensing with the three cytokines, which according to your statement has not been done before. The paper is very well written, easy to follow, well illustrated, and cites relevant up-to-date literature, to put your data into context. Ethics approval for the study is indicated, the methods are sound and well explained, and the data are mostly conclusive and in line with prior studies on PDLSCs and other MSC types. Both, the inflammatory priming approach and readout targets are well chosen and state of the art. Although the results appear to be credible, one minor weakness of the study is that all observed results are strongly depending on the inhibitor/cytokine doses and timing of the approach, which are not yet supported by citing adequate references in the methods and by relating to the doses chosen by others in the discussion. The article would also greatly benefit from providing representative plots for the FACS gating/analysis strategy to see that the method was correctly applied.
Minor Comments:
1) In your introduction, you refer to studies by Wada, Shi and Grontos, (e.g. reference 4 & 16 Wada et al. J Cell Physiol 2009, also reviewed by Wada et al. in a Springer Journal 2015 https://doi.org/10.1007/s40496-015-0062-y) who studied the immunomodulatory properties of PDLSCs and also to reviews (ref 13 and 24) by Krampera et al. and Yufang Shi et al. who were among the pioneers in studying the differential licensing of MSCs by these three cytokines. A key aspect of your article is the quantitative approach, to use defined stimuli (dose and time) and to compare the respective readout. While this is well introduced in the introduction, you do not follow up on the quantitative method in the discussion, but merely discuss your results on a qualitative level. However, the observed differences need be related to the dose of cytokines used in this study and by others and related to each other accordingly, and the dose of both cytokines and inhibitors justified properly by citing adequate references. Line 138 and Line 128-134: For the licensing of the PDLSCs following cytokine concentrations were used: 5 ng/ml of IL1b, 10 ng/ml of TNFa, and 100 ng/ml of IFNg, and considering inhibitors: 50uM PF-06840003 IDO inhibitor, 1uM BMS202 PD-1/PD-L1 interaction inhibitor, 1uM Celecoxib PTGS-2 inhibitor. How are all these chosen concentrations motivated? Please provide and cite adequate references that substantiate the chosen concentrations (e.g. please see review by Yufan Shi Ref 24). Please double-check all these concentrations, since typically 10 ng/ml of either TNFa or IFNg are used, while IFNg is usually expressed in units (activity) and the given value may actually be 100 units/ml not 100 ng/ml, which would be rather high. It would also be strange to compare different cytokines, but use a 10x higher concentration of one over the other. Furthermore titrations should be provided in the supplement.
2) Please provide representative FACS plots to properly outline your gating / analysis strategy in the supplement of this article.
Author Response

(The authors gave the same response as above.)

Round 2
Reviewer 2 Report
The paper is improved.